# SG-Nav: Online 3D Scene Graph Prompting for LLM-based Zero-shot Object Navigation

**Hang Yin**[1][*] **Xiuwei Xu**[1][*][†]**, Zhenyu Wu**[2]**, Jie Zhou**[1]**, Jiwen Lu**[1][‡]
[1]Tsinghua University
[2]Beijing University of Posts and Telecommunications
{yinh23, xxw21}@mails.tsinghua.edu.cn
wuzhenyu@bupt.edu.cn, {jzhou, lujiwen}@tsinghua.edu.cn

## Abstract

In this paper, we propose a new framework for zero-shot object navigation. Existing zero-shot object navigation methods prompt LLM with the text of spatially closed objects, which lacks enough scene context for in-depth reasoning. To better preserve the information of environment and fully exploit the reasoning ability of LLM, we propose to represent the observed scene with 3D scene graph. The scene graph encodes the relationships between objects, groups and rooms with a LLM-friendly structure, for which we design a hierarchical chain-of-thought prompt to help LLM reason the goal location according to scene context by traversing the nodes and edges. Moreover, benefit from the scene graph representation, we further design a re-perception mechanism to empower the object navigation framework with the ability to correct perception error. We conduct extensive experiments on MP3D, HM3D and RoboTHOR environments, where SG-Nav surpasses previous state-of-the-art zero-shot methods by more than **10%** SR on all benchmarks, while the decision process is explainable. To the best of our knowledge, SG-Nav is the first zero-shot method that achieves even higher performance than supervised object navigation methods on the challenging MP3D benchmark. Project page.

## 1 Introduction

Object navigation, which requires an agent to navigate to an object specified by its category in unknown environment [3], is a fundamental problem in a wide range of robotic tasks. Thanks to the advancements in deep learning and reinforcement learning (RL), modular-based object navigation approaches [6; 32; 48] have achieved impressive performance in recent years, which caches the visual perception results of the RGB-D observation into semantic map and learns a policy to predict actions according to this map. However, conventional methods rely on time-consuming training in simulation environment, which only works well on specific datasets and can only handle limited categories of goal object.

In order to solve above limitations and enhance the generalization ability of object navigation, zero-shot object navigation has received increasing attention. In the zero-shot setting, the system does not require any training or finetuning before applied to real-world scenarios, and the goal category can be freely specified by text in an open-vocabulary manner. Since training data is unavailable, recent zero-shot object navigation methods [46; 52] leverage large language models (LLM) [36; 1] instead of RL agent to perform decision-making processes, where the extensive knowledge in LLM helps to address various common-sense reasoning problems in zero-shot. In these works, the LLM is required to select a frontier for the agent to explore the unobserved area. As LLM cannot perceive the

---

[*] Equal contribution. [†] Project lead. [‡] Corresponding author.

38th Conference on Neural Information Processing Systems (NeurIPS 2024).

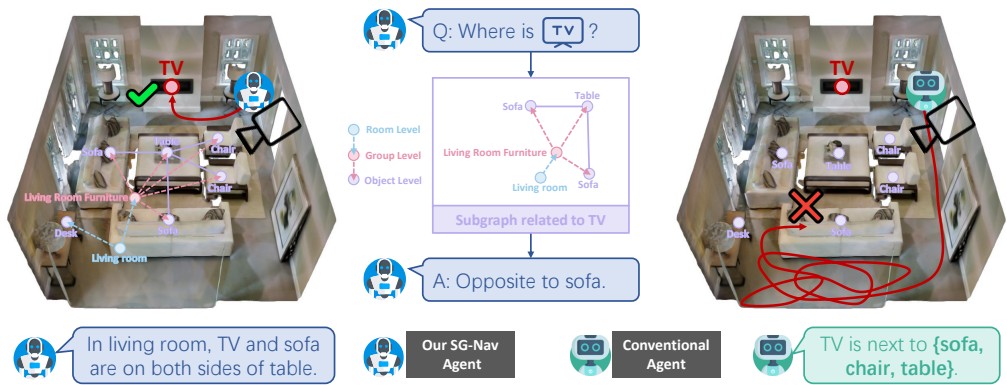

Figure 1: Different from previous zero-shot object navigation methods [46; 52] that directly prompt LLM with text of nearby object categories, we construct a hierarchical 3D scene graph to represent the observed environment and prompts LLM to fully exploit the structure information in the graph. Our SG-Nav preserves fine-grained scene context and makes reasonable and explainable decisions.

environment visually, they select the categories of objects near each frontier to prompt LLM and rate the possibility of each frontier. While these approaches achieve zero-shot object navigation, there are still some drawbacks in their design: (1) they prompt LLM with only the text of nearby object categories, which is less informative and lacks scene context such as spatial relationship; (2) they let LLM directly output the possibility for each frontier, which is an abstract task and does not fully utilizes the reasoning ability of the LLM. As a result, their reasoning process is not explainable and the performance is far from satisfactory.

In this paper, we propose a new object navigation framework namely SG-Nav, which fully exploits the reasoning ability of LLM for zero-shot navigation planning with high accuracy and fast speed. As shown in Figure 1, different from previous LLM-based object navigation framework that prompts LLM with only text of object categories and directly outputs probability, SG-Nav represents the complicated 3D environment in an online updated hierarchical 3D scene graph and utilizes it to hierarchically prompt LLM for reliable and explainable decision making. Benefit from the scene graph representation, we further design a graph-based re-perception mechanism to empower the object navigation framework with the ability to correct perception error. Specifically, we first construct an online 3D hierarchical scene graph along with the exploration of the agent. Since there are many levels and nodes in the scene graph, it is challenging to build the scene graph in real time. To solve this problem, we propose to densely connect the newly detected nodes to previous nodes in an incremental manner. We design a new form of prompt to control the computational complexity of this dense connecting process linear to the number of new nodes, which is followed by a pruning process to discard less informative edges. With the 3D scene graph, we divide it into several subgraphs and propose a hierarchical chain-of-thought to let LLM perceive the structural information in each subgraph for explainable probability prediction. Then we interpolate the probability of each subgraph to the frontiers for decision making, which can be further explained by summarizing the reasoning process for the most related subgraphs. Moreover, we also notice that previous object navigation framework fails when detecting out a false positive goal object. So we propose a graph-based re-perception mechanism to judge the credibility of the observed goal object by accumulating relevant probability of subgraphs. The agent will give up goal object with low credibility for robust navigation. We conduct extensive experiments on MP3D, HM3D and RoboTHOR environments. Our SG-Nav achieves state-of-the-art zero-shot performance on all benchmarks, surpassing previous top-performance zero-shot method by more than 10% in terms of SR. SG-Nav can also explain the reason for its decision at each step, making it more practical in human-agent interaction scenarios.

## 2 Related Works

**Object-goal Navigation:** Objec-goal navigation methods can be divided into two categories: end-to-end RL methods [38; 29; 43; 44; 26; 25; 50] and explicit modular-based methods [6; 32; 48; 11; 46;

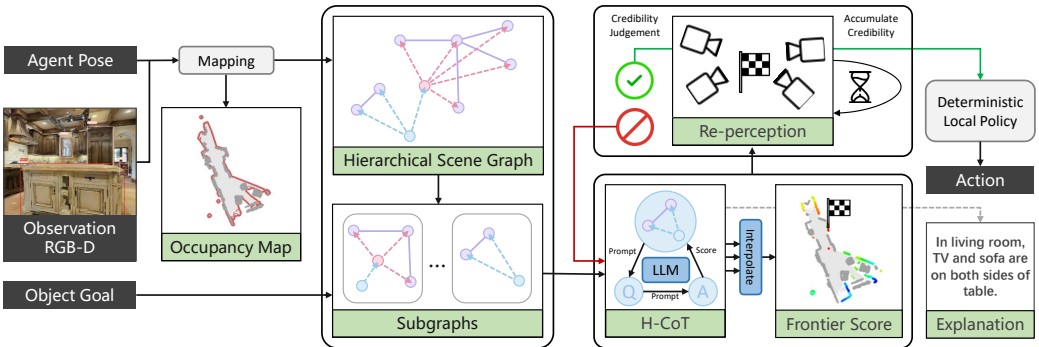

Figure 2: Pipeline of SG-Nav. We construct a hierarchical 3D scene graph as well as an occupancy map online. At each step, we divide the scene graph into several subgraphs, each of which is prompted to LLM with a hierarchical chain-of-thought for structural understanding of the scene context. We interpolate the probability score of each subgraph to the frontiers and select the frontier with highest score for exploration. This decision is also explainable by summarizing the reasoning process of the LLM. With the scene graph representation, we further design a re-perception mechanism, which helps the agent give up false positive goal object by continuous credibility judgement.

52]. DD-PPO [38] implicitly encodes the visual observation into latent code and predicts low-level actions. Follow-up works increase the performance by better visual representation [29; 43], auxiliary tasks [44] and data augmentation [26]. However, since this kind of methods implicitly encode 3D scene and directly predict actions, which may fail to capture fine-grained context. The end-to-end RL also suffers from low sampling efficiency. To solve these problems, modular-based methods map the visual perception results to BEV or 3D map [49; 42; 41] and update it online. The agent learns where to navigate with this semantic map [6; 32; 48]. In persuit of open-vocabulary object navigation, COWs [11] and ZSON [25] align the goal to CLIP embedding [30]. L3MVN [46] and ESC [52] further utilize large language models for zero-shot decision making. Our SG-Nav is also a LLM-based zero-shot framework. But differently, we propose to construct 3D scene graph instead of semantic map to prompt LLM with hierarchical chain-of-thought and graph-based re-perception, which better preserves the scene context and makes the decision more robust.

**Large Pretrained Models for Robotics:** Large pretrained language models [1; 36] and vision-language models [30; 20; 23; 20; 21] have received increasing attention in recent years, which is widely applied in embodied AI tasks including navigation [17; 11; 25; 46; 52; 35; 13; 51], task planning [2; 7; 16; 27; 4; 39; 33; 40], manipulation [22; 14; 15; 47]. It is observed that large vision-language models becomes a good hand in dealing with embodied AI problems due to its end-to-end processing of images and language. NLMap [7] leverages CLIP embedding [30] to build an open-vocabulary map for task planning. Voxposer [15] automatically generates code to interacts with VLMs [28; 18], which produces a sequence of 3D affordance maps and constraint maps to guide the optimization of path planning. However, it is still hard to process streaming RGB-D video for vision-language models. To better make use of the observed 3D scenes, recent methods [52; 51] of navigation describe the scene in pure text and utilize it to prompt LLM. Our SG-Nav, instead, represents the 3D scene using scene graph, which preserves much more context information. The proposed hierarchical chain-of-thought and graph-based re-perception methods can also better exploit the potential of LLM for object navigation.

## 3   Approach

In this section, we first provide the definition of zero-shot object navigation and introduce the overall pipeline of SG-Nav. Then we describe how to construct an open-vocabulary and hierarchical 3D scene graph online. Finally we show how to prompt the LLM with 3D scene graph to achieve context-aware and robust object navigation.

### 3.1 Zero-shot Object Navigation

**Problem Definition.** A mobile agent is tasked with navigating to an object $o$ specified by its category $c$ in an unknown environment. The agent receives streaming RGB-D video as input along with its exploration and decides where to go until it finds the goal object. Formally, at each time instant $t$, the input to the agent is a posed RGB-D image and $o$ and the agent should execute an action $a \sim \mathcal{A}$, where $\mathcal{A}$ consists of `move_forward, turn_left, turn_right` and `stop`. The task is successful if the agent stops within $d_s$ meters of the goal object in less than $T$ steps. Otherwise it fails.

Different from conventional object navigation which requires training in simulation environment with fixed vocabulary of categories $c$, we study zero-shot object navigation: the goal category $c$ can be freely specified with text in an open-vocabulary manner, and the navigation system does not require any training or finetuning, which is of great generalization ability.

**Overview.** Since performing open-vocabulary object navigation without training is very challenging, we resort to large language model (LLM) [36; 1] to solve this problem. LLM demonstrates extensive knowledge and surprising generalization ability, which is a best choice to handle this zero-shot task. To make LLM aware of the visual observation of the environment, we propose to construct an online 3D scene graph, which contains rich context information about the environment and is suitable for interacting with LLM. At each step, we prompt the LLM with our 3D scene graph with hierarchical chain-of-thought and graph-based re-perception for decision making. The pipeline is shown in Figure 2.

### 3.2 Online 3D Scene Graph Construction

In this subsection, we first provide formal definition of our 3D scene graph. Then we detail how to construct the scene graph in real time and online.

**Hierarchical 3D Scene Graph.** A 3D scene graph describe the 3D scene with nodes and edges between these nodes. We define three kinds of nodes: object, group and room, to represent the scenes in different graininess. Here group indicates a set of related objects, such as a dining table and its surrounding chairs, TV and TV cabinet. Edge can exist between any pair of nodes. If two nodes are not in the same level (e,g, chair and living room), the edge between them represents affiliation (i.e. this chair belongs to this living room). If two nodes are in the same level, the edge represents the relationship between them, which can be spatial relationship such as *on the top of* between close objects, or functional relationship between far away objects like a TV and its opposite sofa.

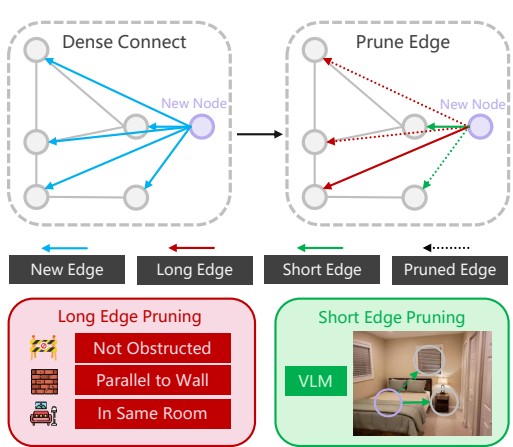

Figure 3: The incremental generation of edges. We densely connect newly registered nodes (purple) to all other nodes by efficiently prompting the LLM. We divide the edges into long edges and short edges and prune less informative ones with different strategies.

**Incremental Updating and Pruning.** Since there are many objects in the environment and the relationship between them are complicated, building the scene graph for the whole scene is time-consuming. To enable real-time online construction of the 3D scene graph, we propose to incrementally update it across frames. That is, at each time $t$, we register new nodes $\mathcal{N}_t$ from the RGB-D observation, and connect these nodes with the previous scene graph $\mathcal{G}_{t-1}$ to acquire $\mathcal{G}_t$.

We adopt different strategy to acquire different level of nodes. For object nodes, we leverage online open-vocabulary 3D instance segmentation method [12] to acquire the 3D instance. Formally, at each time $t$, the method detects and segments objects from the RGB-D observation and synchronously reconstruct the 3D point clouds for the environment. These newly detected objects will be matched with object nodes detected from the previous $t-1$ frames and merged into them. Objects that do not match with any previous one will be registered as a new object node. Each object node has a semantic

category and its confidence score. Then group nodes are computed based on all object nodes up to time $t$. If two object nodes are connected by an edge and their categories are related, we register a group for them. This process will iterate until no new node can be assign to this group. We pre-define a dictionary to tell which pair of categories are related by LLM, which is detailed in Section 6.2. For room nodes, we follow the same paradigm for acquiring object nodes, where room is regarded as instance and thus we can prompt the open-vocabulary 3D instance segmentation model with the room type.

We also design different strategies to build edges between the nodes of different levels. We first connect high-level nodes with their affiliated low-level nodes. For room-object node pair, we think the object belongs to the room if the instance mask of this object on the point clouds is contained in the instance mask of this room. For room-group node pair, we connect them if all objects in this group belongs to the room. We do not additionally connect group and object nodes since the former is registered based on the latter.

Then we connect related object nodes to build intra-level edges, as shown in Figure 3. Note there is no intra-level edges between group and room nodes. For newly registered object nodes at time $t$, we first densely connect them with all object nodes, and then perform pruning to remove less informative edges. We combine $m$ newly registered nodes and $n$ previous nodes to form $m(m+n)$ edges. For each edge, we can prompt LLM with the categories of connected nodes and ask LLM to infer their relationship. However, traversing all $m(m+n)$ edges and feed prompts to LLM requires $\mathcal{O}(m(m+n))$ computational complexity. To speed up the dense connecting process for real-time processing, we propose a new format of prompt: *Predict the most likely relationships between these pairs of objects:* $[\{(a,b)|a \in S_n, b \in S_n \cup S_p\}]$, where $\mathcal{S}_n$ and $\mathcal{S}_p$ are the set of $m$ newly registered nodes and $n$ previous nodes. This enables LLM to generate relationships between all pairs of nodes in one-shot with much less computational cost, for which we provide theoretical derivation below.

According to [37], complexity of processing $L$ tokens for LLM is $\mathcal{O}(L^r)$, where $1 < r \le 2$. Suppose $L_{pro}$ and $L_{res}$ represent the number of tokens of the prompt and LLM's response respectively. Then the computational complexity of processing $m(m+n)$ pairs of nodes for our method and the naive $\mathcal{O}(m(m+n))$ method are $T_{our} = \mathcal{O}\left((L_{pro} + m(m+n) \cdot L_{res} + m(m+n) \cdot 2)^r\right)$ and $T_{naive} = m(m+n) \cdot \mathcal{O}\left((L_{pro} + L_{res})^r\right)$ respectively[2]. Since $L_{pro} \gg 2$ and $L_{pro} \gg L_{res}$, we can approximately treat $\frac{2}{L_{pro}} = 0$ and $\frac{L_{res}}{L_{pro}} = \alpha$ ($\alpha \le 0.01$). Assume that $1 \le m \le 5$, $1 \le n \le 100$. We then compute the ratio between them:

$$\frac{T_{our}}{T_{naive}} = \frac{(L_{pro} + m(m+n) \cdot L_{res} + m(m+n) \cdot 2)^r}{m(m+n) \cdot (L_{pro} + L_{res})^r} < \frac{c}{m+n} \tag{1}$$

where $c$ is a small constant. Consequently, our method reduces the complexity from $\mathcal{O}(m(m+n))$ to $\mathcal{O}(m)$. In this way, the LLM can generate new edge proposals in fast speed.

We further prune the dense connected edges to make the 3D scene graph more precise. We divide newly generated relationships into two categories: long-range and short-range. The basis for this division is whether there is a RGB-D image containing the two connected objects. If true, this edge is short-range, for which we can feed the RGB image to a vision language model (VLM) like LLaVA [23] and ask whether this relationship exists. Nonexistent ones will be pruned. Otherwise the edge is a long-range one. We design two standards to validate it: (1) the line connecting two objects is unobstructed and parallel to the wall of room; (2) these two object nodes belong to the same room node. Only edges meet with both standards can be preserved.

### 3.3 Prompting LLM with 3D Scene Graph

In this subsection, we describe how to prompt LLM with the 3D scene graph at each step for decision making. Since directly predicting $a$ is hard for LLM, we adopt Frontier-based Exploration (FBE) strategy to divide the decision into two part: (1) synchronous with the 3D scene graph, we build an online occupancy map to indicate explored, unexplored and occupied area in BEV, based on which we compute the frontiers of explored area. In order to further explore the environment, the agent must go through one of the frontier. So in high level, the LLM is only required to select one of the frontier which has the highest probability to find $o$; (2) once the frontier to be explore is chosen, we simply apply local policy like Fast Marching Method [34] for path planning and output $a$.

---

[2]We provide detailed proof process and illustrated examples in Section 6.3.

**Hierarchical Chain-of-thought Prompting.** Different from previous methods [46; 52] that prompt LLM with only categories of objects near each frontier, we compute the probability $P^{fro}$ of each frontier by prompting LLM with our 3D scene graph. At time $t$, we divide the scene graph $\mathcal{G}_t$ into subgraphs, each of which is determined by an object node with all its parent nodes and other directly connected object nodes. For each subgraph, we predict $P^{sub}$, the possibility of goal appearing near this subgraph. Then the probability $P_i^{fro}$ of the $i$-th frontier can be averaged by:

$$P_i^{fro} = \sum_{j=1}^{M} \frac{P_j^{sub}}{D_{ij}} \ (i \in \{1, 2, ..., N\}) \tag{2}$$

where $M$ is the total number of subgraphs and $D_{ij}$ is the distance between the center of the $i$-th frontier and the central object node of the $j$-th subgraph.

In order to make the prediction of $P^{sub}$ accurate and explainable, we propose a hierarchical chain-of-thought method to prompt LLM, which fully exploits the structure information contained in the subgraphs. Specifically, for each subgraph, LLM is first prompted with *Predict the most likely distance between* [object] *and* [goal], where [object] means the category of the central object node. Then we prompt LLM to ask question related to the navigation goal, with the prompt: *Ask questions about the* [object] *and* [goal] *for predicting their distance*. After obtaining the questions about goal, we prompt LLM to answer those questions according to the information of subgraph, with the prompt: *Given* [nodes] *and* [edges] *of subgraph, answer above* [questions]. Therefore, from those answers we get all the necessary information required for inferring the position of goal. Finally, we prompt LLM to summarize the conversation and predict $P$, with the prompt *Based on the above conversation, determine the most likely distance between this subgraph and* [goal]. And $P^{sub}$ can be simply computed by taking the inverse of the outputted distance. The LLM also summarizes the analysis process of the nearest 3 subgraphs to the selected frontier to explain the reason for decision.

**Graph-based Re-perception.** Conventional object navigation framework does not take perception error into consideration. Once the agent detects out false positive objects of the goal category, it will not explore or perceive the environment any more and directly navigate to this object.

To solve this problem, we enhance the current object navigation framework with graph-based re-perception mechanism. When the agent detects out a goal object, it will approach to this object, observe it from multiple perspectives and accumulate a credibility score $S$. For the $k$-th RGB-D observation since this moment, we compute the credibility of it by:

$$S_k = C_k \cdot \sum_{j=1}^{M} \frac{P_j^{sub}}{D_j} \tag{3}$$

where $C_k$ is the confidence score of this goal object, $D_j$ is the distance between the central object node of the $j$-th subgraph and this object. The success condition for our re-perception mechanism is:

$$N_{stop} < N_{max}, \ \text{where} \ N_{stop} = N, \ \text{s.t.} \ \sum_{i=1}^{N-1} S_i < S_{thres} \leq \sum_{i=1}^{N} S_i \tag{4}$$

where $N_{max}$ is a pre-defined maximum number of steps. If succeed, the agent will navigate to this goal object without any further exploration or perception. Otherwise, it will give up this object and continue the exploration.

## 4 Experiment

In this section, we first describe our datasets and implementation details. Then we compare our SG-Nav with state-of-the-art object navigation methods in different settings. We further conduct several ablation studies to validate the effectiveness of our design. Finally we demonstrate some qualitative results of SG-Nav.

### 4.1 Benchmarks and Implementation Details

We evaluate our method on three datasets: Matterport3D (MP3D) [5], HM3D [31] and RoboTHOR [9]. MP3D is a large-scale 3D scene dataset, which is used in Habitat ObjectNav challenges. We test our

Table 1: Object-goal navigation results on MP3D, HM3D and RoboTHOR. We compare the SR and SPL of state-of-the-art methods in different settings.

| Method | Unsupervised | Zero-shot | MP3D | | HM3D | | RoboTHOR | |
|---|---|---|---|---|---|---|---|---|
| | | | SR | SPL | SR | SPL | SR | SPL |
| SemEXP [6] | No | No | 36.0 | 14.4 | – | – | – | – |
| PONI [32] | No | No | 31.8 | 12.1 | – | – | – | – |
| ProcTHOR [10] | No | No | – | – | 54.4 | 31.8 | 65.2 | 28.8 |
| ZSON [25] | Yes | No | 15.3 | 4.8 | 25.5 | 12.6 | – | – |
| ProcTHOR-ZS [10] | Yes | No | – | – | 13.2 | 7.7 | 55.0 | 23.7 |
| CoW [11] | Yes | Yes | 7.4 | 3.7 | – | – | 26.7 | 16.9 |
| ESC [52] | Yes | Yes | 28.7 | 14.2 | 39.2 | 22.3 | 38.1 | 22.2 |
| L3MVN [46] | Yes | Yes | 34.9 | 14.5 | 48.7 | 23.0 | 41.2 | 22.5 |
| OpenFMNav [19] | Yes | Yes | 37.2 | 15.7 | 52.5 | 24.1 | 44.1 | 23.3 |
| VLFM [45] | Yes | Yes | 36.2 | 15.9 | 52.4 | **30.3** | 42.3 | 23.0 |
| SG-Nav-LLaMA | Yes | Yes | 40.1 | 16.0 | 53.9 | 24.8 | 47.3 | 23.7 |
| SG-Nav-GPT | Yes | Yes | **40.2** | **16.0** | **54.0** | 24.9 | **47.5** | **24.0** |

SG-Nav on the validation set, which contains 11 indoor scenes, 21 object goal categories and 2195 object-goal navigation episodes. HM3D is used in Habitat 2022 ObjectNav challenge, containing 2000 validation episodes on 20 validation environments with 6 goal object categories. RoboTHOR is used in RoboTHOR 2020, 2021 ObjectNav challenge, containing 1800 validation episodes on 15 validation environments with 12 goal object categories.

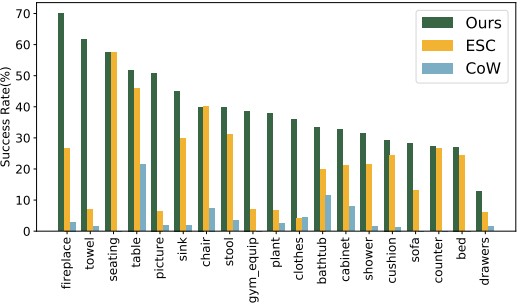

Figure 4: Per category SR on MP3D.

Figure 5: Time cost of connecting $n$ edges.

**Evaluation Metrics:** We report three metrics including *success rate (SR)*, *success rate weighted by path length (SPL)* and *SoftSPL*. SR is the core metric of object-goal navigation task, representing the success rate of successful navigation episode. SPL measures the ability of agent to find the optimal path. If success, SPL $= \frac{\text{optimal path length}}{\text{path length}}$, otherwise SPL $= 0$. SoftSPL [8] measures the navigation progress and navigation efficiency of the agent. Higher is better for all the three metrics.

**Implementation Details:** We set $500$ as the maximal navigation steps. The farthest and closest perceived distances of the agent are $10m$ and $1.5m$. Each step of the agent takes $0.25m$, and each rotation takes $30°$. The camera of agent is $0.90m$ above the ground, and perspective is horizontal. The camera outputs $640 \times 480$ RGB-D images. We maintain a $800 \times 800$ 2D occupancy map with the resolution of $0.05m$, which can represent a $40m \times 40m$ large-scale scene. In equation 4, the hypre-parameters $N_{max}$ is 10 and $S_{thres}$ is 0.8. For the verification of short edge, we adopt LLaVA-1.6 (Mistral-7B) [24] as the VLM for discrimination. We choose LLaMA-7B [36] and GPT-4-0613 as the LLM for our SG-Nav, which are discriminated by SG-Nav-LLaMA and SG-Nav-GPT.

## 4.2 Comparison with State-of-the-art

We compare SG-Nav and with state-of-the-art object navigation methods of different setting, including supervised, unsupervised and zero-shot methods in Table 1. SG-Nav surpasses previous zero-shot

Table 2: Effect of the 3D scene graph (SG) and re-perception mechanism (RP).

| Method | MP3D | | | HM3D | | | RoboTHOR | |
|---|---|---|---|---|---|---|---|---|
| | SR | SPL | SoftSPL | SR | SPL | SoftSPL | SR | SPL |
| SG-Nav w/o SG&RP | 25.7 | 12.9 | 22.6 | 38.6 | 18.9 | 27.7 | 34.9 | 21.1 |
| SG-Nav w/o RP | 36.5 | 15.0 | 23.9 | 49.6 | 23.6 | 33.0 | 41.9 | 22.6 |
| SG-Nav | **40.1** | **16.0** | **24.9** | **53.9** | **24.8** | **33.8** | **47.3** | **23.7** |

methods by about 10% on all three benchmarks. We also notice that on the challenging MP3D dataset, SG-Nav even outperforms supervised methods including SemEXP and PONI. Note that the improvement of SR is larger than SPL, this is because our graph-based re-perception mechanism can help the agent give up false positive goal object, which can significantly increase the success rate. However, during the process the re-perception, the agent still needs to approach to the goal to observe it from multi views. So this strategy cannot reduce the length of path.

We also compare per-category success rate of different zero-shot methods in Figure 4. For all goal categories, SG-Nav outperforms other methods by a large margin, especially on categories that have more relationships with other objects such as fireplace and towel.

Table 3: Effect of the room level node and group level node on MP3D dataset.

| Room | Group | SR | SPL | SoftSPL |
|---|---|---|---|---|
| × | × | 38.3 | 15.3 | 23.9 |
| ✓ | × | 39.0 | 15.8 | 24.4 |
| × | ✓ | 39.4 | 15.8 | 24.6 |
| ✓ | ✓ | **40.1** | **16.0** | **24.9** |

Table 4: Effects of the edges in scene graph $\mathcal{G}$ on MP3D dataset.

| Method | SR | SPL |
|---|---|---|
| $\mathcal{G}$ with only nodes | 38.0 | 15.2 |
| Remove short edges from $\mathcal{G}$ | 39.0 | 15.5 |
| Remove long edges from $\mathcal{G}$ | 39.5 | 15.7 |
| **Complete $\mathcal{G}$** | **40.1** | **16.0** |

## 4.3 Ablation Study

**Scene Graph and Re-perception.** In Table 2, we compare the design choices of scene graph (SG) and re-perception (RP). Without RP, SG-Nav will move to the goal once the detector finds a goal, even thought the goal is a false positive. Due to the complexity of the scene, it is common for detector to make mistakes, which leads to irreversible navigation. If we further remove the whole SG, our framework degenerates to a random exploration model. The agent randomly selects frontier, resulting in a significant performance degradation on all datasets.

Table 5: Effects of the CoT prompting on MP3D dataset.

| Method | SR | SPL |
|---|---|---|
| Text prompting | 36.5 | 14.9 |
| Text prompting seperately | 37.0 | 15.0 |
| w/o Inter-level Edges $\mathcal{G}$ | 38.2 | 15.5 |
| **Raw** | **40.2** | **16.0** |

**Room Node and Group Node.** As shown in Table 3, we compare the performance between different architectures of scene graph by ablating the design of group and room nodes. We remove the room or group node with the edges connected to it and obtain 4 different scene graph architecture. The removal of room node and group node results in 0.7% and 1.1% SR degradation on MP3D dataset respectively. We observe the group node is more important than room node, which is because the group nodes aggregate objects into a whole and thus reduce the redundant information and the complexity of scene graph.

**Efficiency of Scene Graph Construction.** In Figure 5, we show the speed of generating densely connected new edges for our proposed efficient prompt-based method and the naive $\mathcal{O}(m(m+n))$ method. The naive method requires time consumption that is linear to the number of edges, while our efficient prompt significantly reduces the computational complexity.

**Effects of edges.** In Table 4, we study the importance of different kinds of edges. The results validate that edges play an important role in our framework, which helps the hierarchical chain-of-thought better exploit the structural information contained in the environment.

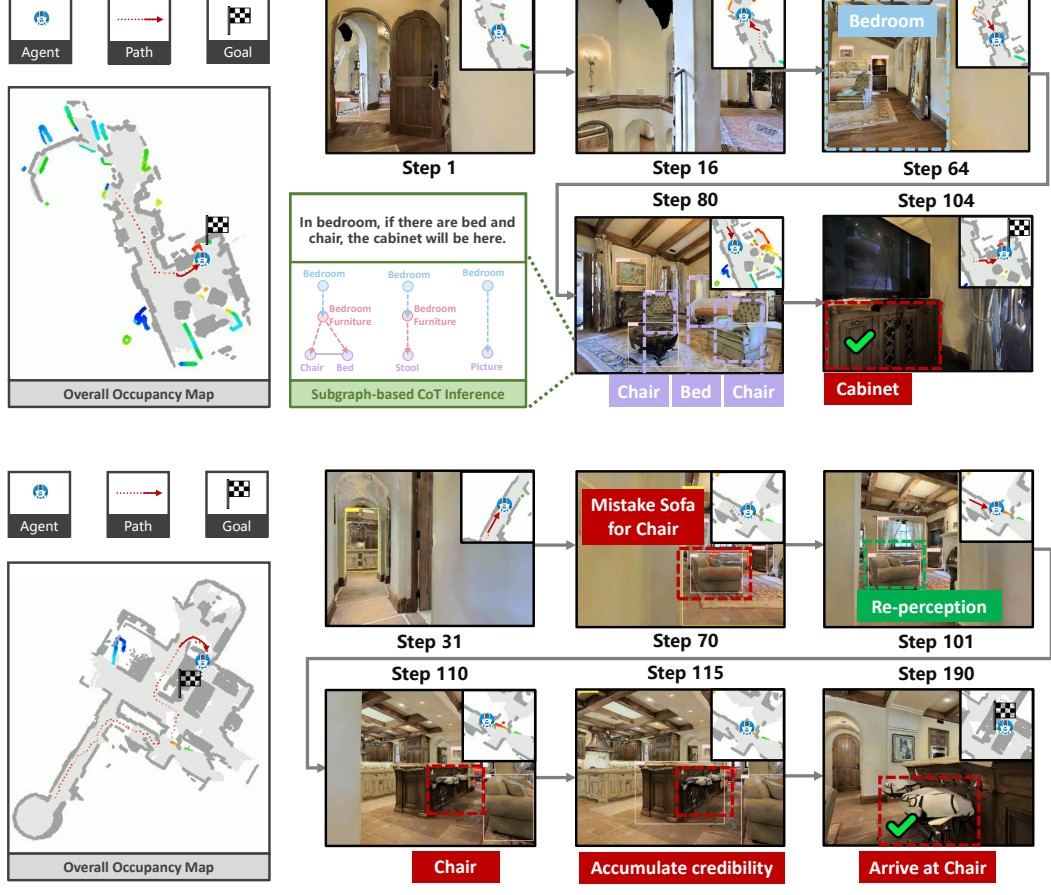

Figure 6: Visualization of the navigation process of SG-Nav.

**Effects of CoT.** In Table 5, we demonstrate the importance of CoT prompting. "Text prompting" refers to directly converting the 3D scene graph into text and prompting the LLM. "Text prompt separately" means converting the nodes and edges into text individually. Additionally, we removed the edges between layers to further validate the necessity of hierarchical graph structure. The experimental results show that our CoT prompting can effectively exploit the structure information contained in the 3D scene graph.

### 4.4 Qualitative Analysis

We visualize the navigation process of SG-Nav in Figure 6. The upper example shows that SG-Nav is able to explain the reason for its decision based on the nearest 3 subgraphs to the selected frontier. The lower example demonstrates that SG-Nav can accumulate credibility of the detected goal object and utilize it to judge the authenticity of this goal, which effectively corrects perception error.

## 5 Conclusion

In this paper, we have proposed SG-Nav, a zero-shot object-goal navigation framework that prompts LLM with scene graph to inference the position of goal object. We first construct a hierarchical 3D scene graph online to preserve rich context information of the environment. To speed up the building of scene graph, we propose an incremental updating and pruning strategy, which first densely connects newly registered nodes to all nodes by efficiently prompting the LLM. Then we divide the edges into long-range and short-range to prune them with different strategies. At each step, we divide the scene graph into subgraphs and prompt LLM with hierarchical chain-of-thought to

output the probability score of each subgraph. Then the score for each frontier can be acquired by simple interpolation, which is also explainable as we can summarize the decision process of LLM on each subgraph. With the graph representation, we further design a re-perception mechanism to empower SG-Nav with the ability to give up false positive goal objects and thus correct perception error. Extensive experiments on three datasets validate the great performance of SG-Nav and the effectiveness of our design.

**Potential Limitations.** Despite of the great performance, SG-Nav still has several limitations: (1) SG-Nav relies on online 3D instance segmentation method for scene graph construction. Currently we adopt 2D vision-language model and vision foundation model to segment each frame and merge the masks across different frames, which is not end-to-end and 3D-aware. A stronger online 3D instance segmentation model will further boost the performance of SG-Nav. (2) SG-Nav can only handle object-goal navigation yet. However, due to the strong zero-shot generalization ability of LLM and the rich context contained in 3D scene graph, we believe SG-Nav can be further extended to more tasks such as image-goal navigation and vision-and-language navigation.

## Acknowledgments

This work was supported in part by the National Natural Science Foundation of China under Grant 62125603, Grant 62336004, and Grant 62321005, and in part by the Beijing Natural Science Foundation under Grant L247009.

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

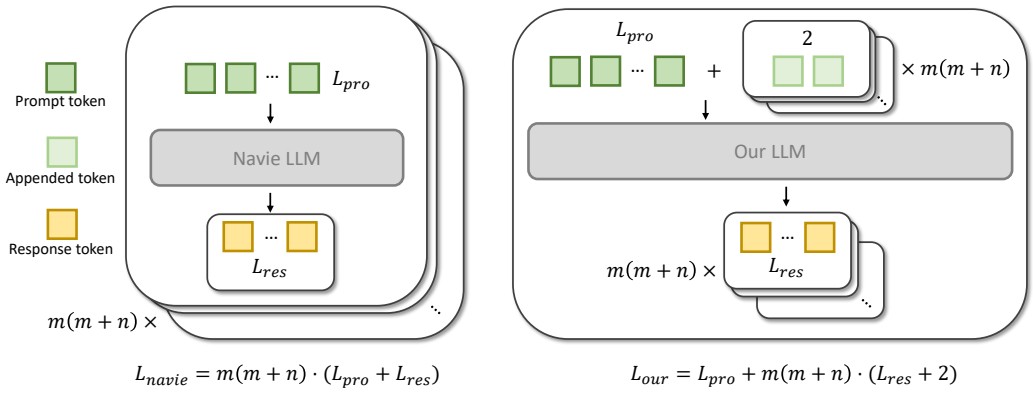

$$L_{navie} = m(m+n) \cdot (L_{pro} + L_{res})$$

$$L_{our} = L_{pro} + m(m+n) \cdot (L_{res} + 2)$$

Figure 7: The illustration of the computational complexity.

# 6 Appendix

## 6.1 Overview

In this appendix, we first provide the dictionary for grouping object nodes in Section 6.2. Then we provide the proof process and illustrated example on computational complexity of our efficient edge generation method in Section 6.3. Next we provide the complete prompts used in our approach in Section 6.4. Finally we provide more visualization results in Section 6.5.

## 6.2 Related Categories

When registering a new group node, we need to judge whether two object categories are related. We use LLM to generate the relationship dictionary, as shown below:

*(Bed, Nightstand) (Wardrobe, Dresser) (Bookshelf, Chair) (Counter, Stove) (Table, Chair) (Bathroom Sink, Mirror) (Shower, Bathtub) (Refrigerator, Freezer) (Oven, Microwave) (Washing Machine, Dryer) (Sofa, Table) (Desk, Office Chair) (Computer, Monitor) (Piano, Bench) (Fireplace, Mantel) (Table, Mirror) (Window, Curtains) (Closet, Hangers) (Bathroom Cabinet, Toiletries) (Living Room Rug, Coffee Table) (Kitchen Cabinet, Dishes) (Dining Room Chandelier, Dining Table) (Clock, Wall) (Floor Lamp, Reading Chair) (Couch, Throw Pillows) (Bookcase, Books)*

## 6.3 Time Complexity for Edge Updating

Here we provide detailed proof for Eq (1). As shown in Figure 7, the computational complexity of processing $m(m+n)$ pairs of nodes for the naive method is $\mathcal{O}(m(m+n))$ and the time is $T_{naive} = m(m+n) \cdot \mathcal{O}((L_{pro} + L_{res})^r)$. Complexity of our method is $T_{our} = \mathcal{O}((L_{pro} + m(m+n) \cdot L_{res} + m(m+n) \cdot 2)^r)$, where $L_{pro} \gg 2$ and $L_{pro} \gg L_{res}$.

So we can approximate that $\frac{2}{L_{pro}} = 0$ and $\frac{L_{res}}{L_{pro}} = \alpha$ ($\alpha \le 0.01$). According to the actual scene graph, usually $1 \le m \le 5$, $1 \le n \le 100$. Then we compute the ratio as:

$$\frac{T_{our}}{T_{naive}} = \frac{(L_{pro} + m(m+n) \cdot L_{res} + m(m+n) \cdot 2)^r}{m(m+n) \cdot (L_{pro} + L_{res})^r} = \frac{\left(1 + \frac{m(m+n) \cdot L_{res}}{L_{pro}} + \frac{m(m+n) \cdot 2}{L_{pro}}\right)^r}{m(m+n) \cdot \left(1 + \frac{L_{res}}{L_{pro}}\right)^r} \quad (5)$$

Since $\frac{2}{L_{pro}} = 0$, we have:

$$\frac{T_{our}}{T_{naive}} = \frac{1}{m(m+n)} \cdot \left(\frac{1 + (m^2 + mn) \cdot \alpha}{1 + \alpha}\right)^r \quad (6)$$

Since $\alpha$ meets $0 < \alpha \ll 1$, we can use the small value approximation principle $(\frac{1+p\alpha}{1+q\alpha})^r \approx (1 + (p - q)\alpha)^r$ to simplify the equation:

$$\frac{T_{\text{our}}}{T_{\text{naive}}} < \frac{1}{m+n} \cdot \frac{\left(1 + \left(m^2 + mn - 1\right) \cdot \alpha\right)^r}{m} \tag{7}$$

Then we use the prior $1 \le m \le 5$ and $1 \le n \le 100$ to scale the coefficient into a constant $c$. The coefficient monotonically increases with $r$, $n$ and $\alpha$:

$$\frac{\left(1 + \left(m^2 + mn - 1\right) \cdot \alpha\right)^r}{m} \overset{r=2, n=100, \alpha=0.01}{\le} \frac{\left(0.01m^2 + m - 0.01\right)^2}{m} \tag{8}$$

The coefficient decreases first and then increases with $m$. It takes the maximum value when $m$ is at its maximum:

$$\frac{\left(0.01m^2 + m - 0.01\right)^2}{m} \overset{m=5}{\le} 5.49 \tag{9}$$

Consequently, we obtain the ratio:

$$\frac{T_{\text{our}}}{T_{\text{naive}}} < \frac{c}{m+n} \tag{10}$$

where $c = 5.49$. Our method reduces the complexity by $(m + n)$ times, from $\mathcal{O}(m(m + n))$ to $\mathcal{O}(m)$. So LLM generates relationship with $\mathcal{O}(m)$ complexity, which is much faster than the naive baseline.

## 6.4 Prompt

We provide the complete prompts of SG-Nav.

**Edge Connecting.** In the edge connecting process, the complete prompt is as follows:

```
You are an AI assistant with commonsense and strong ability to infer the
spatial relationships in an indoor scene.

You need to provide the possible spatial relationships between several
pairs of objects. Relationships include ["next to", "above", "opposite
to", "below", "inside", "behind", "in front of", ...].

All the pairs of objects are provided in JSON format, and you should
also response in JSON format. Here are 2 examples:

1.
Input:
[{"object1": "chair", "object2": "table"}, {"object1": "monitor",
"object2": "desk"}]
Response:
[{"relationships": "next to"}, {"relationships": "above"}]

2.
Input:
[{"object1": "sofa", "object2": "TV"}, {"object1": "plant", "object2":
"chair"}]
Response:
[{"relationships": "opposite to"}, {"relationships": "behind"}]

Now you predict these pairs of objects: [{"object1": {}, "object2":
{}}, {"object1": {}, "object2": {}}, ...]
```

where {} will be replace by the object categories.

**Hierarchical CoT.** In the process of goal inference, we use the following prompts to let LLM predict the most likely position of goal.

**Step 1 (Predict Distance of Object):**

```
You are an AI assistant with commonsense and strong ability to infer the
distance between two objects in an indoor scene.

You need to predict the most likely distance of two objects in a room.
You need to answer the distance in meters and give your reason.  Here is
the JSON format:

Input:

{"object1":  "table", "object2":  "chair"}

Response:

{"distance":  0.5, "reason":  "Because there is always a chair next to
the table."}

Now predict the distance and give your reason:  {"object1":  {},
"object2":  {}}
```

where {} will be replace by the object.

**Step 2 (Ask Question):**

```
You are an AI assistant with commonsense and strong ability to infer the
spatial relationships in an indoor scene.

But you have insufficient information.  You need to ask a question
about the spatial relationship between the object and the goal in the
following JSON format:

Input:

{"object":  "sofa", "goal":  "TV"}

Response:

{"question":  "Is there a table next to the sofa?"}

Now ask question:  {"object":  {}, "goal":  {}}
```

where {} will be replace by the object and goal.

**Step 3 (Answer Question):**

```
You are an AI assistant with commonsense and strong ability to answer
question about the objects in an indoor scene.

Given a graph scene of the scene, you need to answer the question in the
following JSON format:

Input:

{"subgraph":  {"nodes":  ["sofa", "table", ...], "edges":  ["sofa next to
table", ...]}, "question":  "Is there a table next to the sofa?"}

Response:

{"answer":  "Yes"}

Now answer question:  {"subgraph":  {}, "question":  {}}
```

where {} will be replace by the subgraph and question.

**Step 4 (Predict Distance of Subgraph):**

```
You are an AI assistant with commonsense and strong ability to infer the
distance between a subgraph and a goal in an indoor scene.

You need to predict the most likely distance of a subgraph and a goal in
a room.  You need to answer the distance in meters and give your reason.
Here is the JSON format:

Input:

{"subgraph": {"nodes": ["sofa", "table", ...], "edges": ["sofa next to
table", ...]}, "goal":  "TV"}

Response:

{"distance":  2, "reason":  "Because TV and sofa are on both sides of
table."}

Now predict the distance and give your reason:  {"subgraph":  {}, "goal":
{}}
```

where {} will be replace by the subgraph and goal.

## 6.5 Visualization

We further provide visualizations on the constructed scene graphs, the frontier scores along with the
LLM reasoning output, and failure case to help comprehensive understanding of our approach.

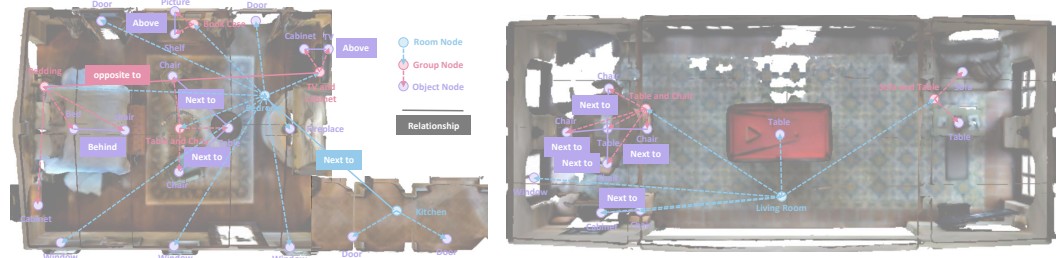

Figure 8: Visualization of two scene graphs.

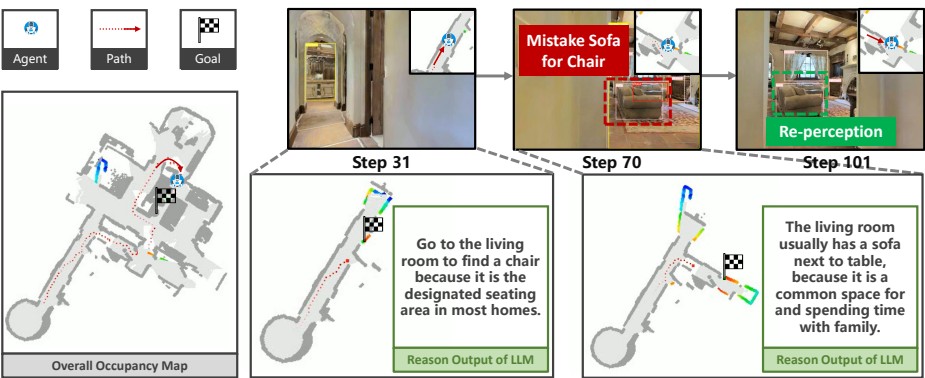

Figure 9: Visualization for the sequential frontier scoring and reasoning output of LLM. We color the
frontiers, where frontiers with high score are in red and ones with low score are in blue.

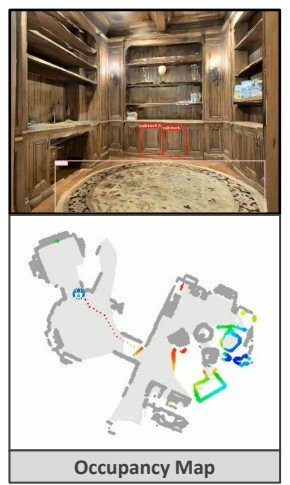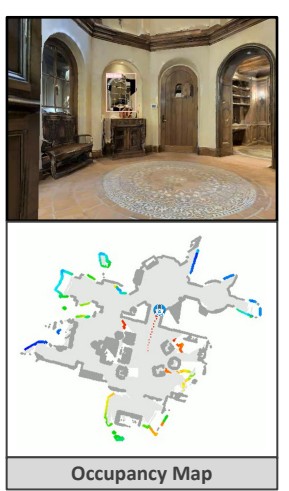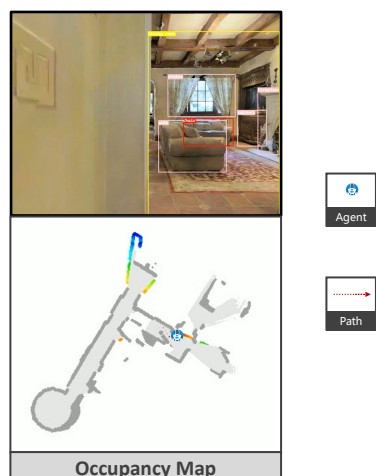

Figure 10: Three failure cases of navigation. In the first case, agent mistakes shelf for cabinet and navigates to it. In the second case, agent fails to detect the goal object. In the third case, LLM predicts a wrong location of chair because there are sofa and TV near this frontier, where there is high probability of chair to appear here. This makes the agent harder to find the goal in limited steps.

