# OpenReview forum: "SG-Nav: Online 3D Scene Graph Prompting for LLM-based Zero-shot Object Navigation"
_NeurIPS.cc/2024/Conference — NeurIPS 2024 poster_

### Official Review · Reviewer_JsEs · 2024-07-03

**Soundness:** 2
**Presentation:** 3
**Contribution:** 2
**Rating:** 3
**Confidence:** 4

**Summary:**

This paper presents a new framework for zero-shot object navigation. Unlike previous methods that only provide objects in close proximity, this paper constructs a scene graph that captures the relationships between objects, groups, and rooms.
This scene graph allows for a more comprehensive understanding of the environment by the navigation system, enabling it to make decisions based on a wider context rather than immediate surroundings. This method significantly improves the robustness and versatility of the navigation system, making it more effective in a variety of settings.

**Strengths:**

1. The paper tries to solve the problem of zero-shot navigation with LLM and scene graph, and experimental results show that this combination does improve the navigation performance.

2. The way the scene graph is updated is interesting. The authors combine the capabilities of LLM and VLM to create connections between nodes and reduce irrelevant edges to make the graph more precise.

3. The experimental results show that there is no significant difference between the results of LLaMA-7B and GPT-4, indicating that the method can work in small-volume LLMs, which is friendly to the demand of computational resources.

**Weaknesses:**

1. The use of scene graphs is not very new in the field of navigation. I think it's important to discuss the use of the scene graph in navigation in related work.

2. I think the authors should take some LLM-based object Navigation method as baseline in the experiments to compare with the proposed method. The authors mention in the abstract that the previous LLM based navigation method promotes LLM with the text of spatially closed objects. Yet no comparison is made in the experimental section, which hardly demonstrates the superiority of the scene context. For example, "L3MVN: Leveraging Large Language Models for Visual Target Navigation" shows comparable success rate than the proposed method in HM3D benchmark, which is also mentioned in the Line 77.

3. The way LLM utilizes the scene graph is still through text prompt, which makes me think that the authors have overpacked it as "LLM friendly structure" a little bit. I think here the usage of the scene graph is just a prompt design and doesn't reflect the hierarchy of the scene graph very well. I consider this reference to be the basis on which I made this judgment: "SayPlan: Grounding Large Language Models using 3D Scene Graphs for Scalable Robot Task Planning".

**Questions:**

1. Line 198， I think a hint that $a$ is an action would be clearer. In addition, why is it said that direct prediction of action by LLM is difficult? Any references or experiments to prove this?

**Limitations:**

The authors discuss limitation very well.

---

> ### Author Rebuttal · Authors · 2024-08-06
>
> **1. Discussion of scene-graph-based navigation**
>
> We thank the reviewer for the constructive advice. We will add more literature on scene graph-based navigation in the final version of paper.
> 3D scene graph is widely utilized in various embodied tasks, such as grounding [1] and task planning [2]. In the field of navigation, we find [3,4] most relevant to our work, which focus on graph-based ObjectNav.
> StructNav [3] constructs a scene graph to represent the observed scene. It adopts BERT to predict the similarity between each object node and the goal. Then the scene graph is used to propagate semantics to the geometric frontiers for action planning. Here the edges of scene graph are only used to represent connectivity, rather than relationship between objects.
> VoroNav [4] constructs a Voronoi graph, where nodes represent regions and edges represent whether two regions are connected. VoroNav directly prompts LLM with the object categories contained in each node to score regions for navigation. The graph is only used for determining explorable area.
> These two methods do not make use of the relationship between objects. On the contrary, SG-Nav constructs a hierarchical 3D scene graph with rich node and edge information. We also propose a Chain-of-Thought Prompting method to exploit the nodes and edges contained in scene graph with LLM. In this way, LLM can understand the hierarchical and structural information in the 3D scene graph for better reasoning.
>
> [1] Context-Aware Entity Grounding with Open-Vocabulary 3D Scene Graphs, CoRL 2023
>
> [2] SayPlan: Grounding Large Language Models using 3D Scene Graphs for Scalable Robot Task Planning, CoRL 2023
>
> [3] How To Not Train Your Dragon: Training-free Embodied Object Goal Navigation with Semantic Frontiers, RSS 2023
>
> [4] VoroNav: Voronoi-based Zero-shot Object Navigation with Large Language Model, ICML 2024
>
> **2. Comparison with more LLM-based navigation methods**
>
> We have compared with ESC, a LLM-based ObjNav method in our experiments. Following the reviewers’ advice, we further compare with some more recent LLM-based methods including L3MVN, VoroNav and OpenFMNav on MP3D, HM3D and RoboTHOR. We uniformly set LLaMA-7B as the LLM for fair comparison. The results are shown in the table below. It is shown that SG-Nav achieves state-of-the-art performance on the challenging MP3D and RoboTHOR benchmarks. On HM3D, SG-Nav also shows comparable performance with the most recent methods. We observe SG-Nav works better on MP3D and RoboTHOR than on HM3D. This is because HM3D is the relatively simplest benchmark among the three, where SG-Nav’s ability to reason and plan in complex and large scenes cannot be fully utilized. Considering the overall performance on three benchmarks, SG-Nav shows leading advantage, especially on complicated and challenging scenarios.
>
> | **Method** | **MP3D SR** | **MP3D SPL** | **HM3D SR** | **HM3D SPL** | **RoboTHOR SR** | **RoboTHOR SPL** |
> | -- | :--: | :--: | :--: | :--: | :--: | :--: |
> | L3MVN | 34.9 | 14.5 | 48.7 | 23.0 | 41.2 | 22.5 |
> | VoroNav | 31.2 | 14.2 | 42.0 | **26.0** | 38.4 | 22.2 |
> | OpenFMNav | 37.2 | 15.7 | **52.5** | 24.1 | 44.1 | 23.3 |
> | **SG-Nav** | **40.2** | **16.0** | 49.1 | 24.3 | **47.5** | **24.0** |
>
> [1] L3MVN: Leveraging Large Language Models for Visual Target Navigation, IROS 2023
>
> [2] VoroNav: Voronoi-based Zero-shot Object Navigation with Large Language Model, ICML 2024
>
> [3] OpenFMNav: Towards Open-Set Zero-Shot Object Navigation via Vision-Language Foundation Models, NAACL 2024
>
> **3. The usage of hierarchy in SG-Nav**
>
> We have considered the hierarchical graph structure in our Chain-of-Thought Prompting. Although we prompt LLM with only texts, the LLM is required to exploit the graph through nodes and their connected edges. For example, the prompt “Given [nodes] and [edges] of subgraph, answer above [questions].” makes the reasoning of LLM aware of the graph structure.
> The ablation study in Table 4 in our paper also shows that there will be a performance drop if we remove all edges and keep only nodes. This means our method really makes use of the graph structure.
> We further conduct ablation experiments by removing edges between nodes in different level. We observe the performance on MP3D changes from **40.2/16.0** to **38.2/15.5** (SR/SPL), which shows that our CoT-Prompting effectively utilize the hierarchical information of the 3D scene graph.
> But we agree with the reviewer that how to better exploit the structure information contained in 3D scene graph with LLM is a very promising direction. The current prompting strategy could be further improved (e.g., by designing a more graph-related workflow for LLM), which we leave for future work.
> | **Method** | **SR** | **SPL** |
> | -- | :--: | :--: |
> | Removing Inter-level Edges | 38.2 | 15.5 |
> | **Raw** | **40.2** | **16.0** |
>
> **4. Explanation of $a$ in Line 198**
>
> We have defined $a$ in L105, which means the action of agent. It is very difficult to directly predict action for navigation. In most state-of-the-art object navigation methods [1,2,3], the prediction is divided into global and local policy. First, a global policy is adopted to predict the position to be explored. Then a local policy is used to plan action based on the predicted position and occupancy map. As for LLM, its advantage mainly lies in reasoning, so it is more suitable for LLM to serve as a global policy, while action planning can be simply solved by traditional methods such as Fast Marching Method and A*.
>
> [1] PONI: Potential Functions for ObjectGoal Navigation with Interaction-free Learning, CVPR 2022
>
> [2] L3MVN: Leveraging Large Language Models for Visual Target Navigation, IROS 2023
>
> [3] ESC: Exploration with Soft Commonsense Constraints for Zero-shot Object Navigation, ICML 2023

---

> > ### Comment · Reviewer_JsEs · 2024-08-12
> >
> > Thanks to the author for the reply. I appreciate the author's detailed discussion of the scene graph based navigation approach. Here are my questions.
> >
> > On the HM3D website(https://aihabitat.org/datasets/hm3d/), the authors of HM3D compare in detail the differences between MP3D, RoboTHOR and HM3D. As we can see from the table, HM3D is not the "simplest" as the authors claimed.
> >
> > I am grateful to the authors for their detailed comparison of other LLM-based approaches. But this raises more questions. For a fair comparison, the authors performed all methods with LLaMA-7B. **I think this is fair but not reasonable.** As seen in the paper, the performance of the authors' method under GPT-4 and under LLaMA-7B is very similar (almost identical). This means that LLM's improved reasoning is hardly a gain for the authors' methods. However GPT-4 or other stronger open source LLMs are not inaccessible in real world. In a real deployment, it is unlikely that users would be able to forcibly restrict the model used for navigation to LLaMA-7B, and they would be perfectly reasonable to use a stronger model for stronger performance - but the authors' methods are unable to show that stronger LLMs result in stronger performance. But for baselines, for example, OpenFMNav achieves better results with GPT-4 and is superior to the author's SG-Nav-GPT-4 in HM3D.
> > Moreover, the authors of L3MVN used the RoBERTa-large model and achieved better results than in the table shown by the authors and is superior to the author's SG-Nav-GPT-4 in HM3D. I think it is unreasonable for the authors to use LLaMA-7B instead of RoBERTa-large because the two are not pre-trained in the same way probably not in line with the motivation of the authors of L3MVN. And the number of RoBERTa-large parameters is much smaller than that of LLaMA-7B.
> > In summary, the authors did not adequately test the potential of all methods under reasonable conditions. So I don't think the author's comparison is reasonable.
> >
> > I thank the authors for the additional experiments on The usage of hierarchy in SG-Nav. One of my concern is that building a scene graph relies heavily on LLM's predictions (Line 172-176), and using a scene graph relies heavliy on LLM's predictions (Line 213-223). With LLM highly involved in navigation, theoretically the navigation performance should be affected by the reasoning ability of LLM, however, the authors show that the experiments with GPT and LLaMA-7B show that the navigation performance is almost independent of the reasoning ability of LLM, can the authors do some analysis on this? Why do GPT-4 and LLaMA behave almost indistinguishably in the navigation of this graph structure?
> >
> > My other big concern is that even if scene structure is used as part of the prompts, the novelty is limited as the method is largely a prompt engineering. I consider the authurs' method lacking in novelty, in large part because of recent new developments in the integration between LLM and scene graph. For example, in SayPlan, the LLM can choose to collapse nodes in the scene graph that are irrelevant to the task and unfold nodes that are relevant.
> >
> > In short, the authors' use of graph structure as prompts for LLM lacks novelty and has some limitations. These limitations suggest that the authors' method is unable to obtain navigation performance improvements with LLM performance improvements. Therefore, the authors' method cannot exceed the performance of the same type of method when both use GPT-4. I think these limitations are a big disadvantage, because nowadays, with the development of LLM, there are more and more models with smaller size and more reasoning power, which other methods can benefit from, but the author's method cannot. Taking all these considerations into account, I keep my rating.

---

> > > ### Author Response · Authors · 2024-08-12
> > >
> > > We thank the reviewer for the reply. The reviewer still have two concerns, for which we provide our answers as below. (With less than two days to go, we're not providing experimental results.)
> > >
> > > **1. SG-Nav is unable to obtain performance boost when using a stronger LLM**
> > >
> > > The reviewer may think that the reason why SG-Nav achieves similar performance when using Llama-7B and GPT-4 resepctively is due to our method cannot fully exploit the reasoning ability of LLM. However, we should point out that this view is not correct. Other state-of-the-art LLM-based method, like OpenFMNav, prompts LLM with very long and complex scene context. In this way, LLM needs to analyze the complex and unstructured text, which requires very strong reasoning and analyzing ability. So using GPT-4 will lead to better performance that Llama-7B. While in our SG-Nav, we prompt the LLM with the subgraph by using chain-of-thought. In this way, LLM only needs to understand the structure information contained in 3D scene graph and follow the proposed sub-reasoning processes, which is a much easier task with more formatted input. With our chain-of-thought promting, even Llama-7B is able to fully understand the scene context and achieve comparable performance with GPT-4. In future work, **we can construct more detailed and larger 3D scene graph to prompt LLM**. In this case, smaller LLM may not have the ability to fully exploit the large 3D scene graph, and LLM with stronger reasoning ability will achieve better performance.
> > >
> > > Moreover, we should highlight that **the ability to achieve high performance with only Llama-7B is a huge advantage of our method**. Just as the reviewer comments in the strength part: “The experimental results show that there is no significant difference between the results of LLaMA-7B and GPT-4, indicating that the method can work in small-volume LLMs, which is friendly to the demand of computational resources.” Note that in real world application, the navigation framework may need to be deployed on the robot agent. In this case, using GPT-4 as the LLM will be very slow and expensive. On the contrary, Llama is open-sourced and there are many works focusing on how to improve the efficiency of Llama for better deployment. So we think Llama-7B will be a good choice for real robots.
> > >
> > > **2. Lack of novelty**
> > >
> > > The reviewer thinks SG-Nav lacks novelty because recent works such as SayPlan conducts in-depth study on the integration between LLM and scene graph. However, we think this judgement is unfair.
> > >
> > > First, **SayPlan studies a totally different task with ours**. In their work, they assume a pre-constructed 3D scene graph is avaible and mainly focus on task planning. While in SG-Nav, we propose to build an online 3D scene graph to prompt LLM for navigation reasoning.
> > >
> > > Second, **the novelty of SG-Nav includes an efficient online 3D scene graph construction method, a hierarchical chain-of-thought prompting method and a re-perception mechanism**. We propose an online 3D scene graph-based navigation framework, rather than just focusing on how to let LLM understand a given 3D scene graph. In terms of the chain-of-thought prompting, we have explained its novelty in the rebuttal. We also notice that reviewer zzdn thinks our chain-of-thought prompting is interesting.

---

> > > > ### Comment · Reviewer_JsEs · 2024-08-12
> > > >
> > > > I appreciate the authors' responses. But I still have some concerns. I'll try to respond to authors' reply in as timely as I can.
> > > >
> > > > The authors claim that "LLM only needs to understand the structure information contained in 3D scene graph and follow the proposed sub-reasoning processes, which is a much easier task with more formatted input." However, current LLMs are very capable of reasoning and do not require the authors to intentionally make reasoning less difficult. The biggest limitation of SG-Nav is that navigation performance cannot be improved when LLM reasoning performance is improved; while other methods can be improved. I think this limitation is significant because LLM options can be more flexible when it comes to real-world deployments. I agree with the authors that GPT-4 is slow and expensive when deployed in real environments; therefore small volume LLM is something that should be prioritized. However, LLaMA-7B is not the only small-volume LLM; considering the timing of the release and the NeurIPS submission deadline, I think LLaMA-2-7B and LLaMA-3-8B should also be considered. These two LLMs are stronger than LLaMA-7B. With these small-volume models with stronger reasoning, the performance of other methods is improved, but not SG-Nav. From this perspective, SG-Nav does not have a performance advantage over other methods of the same type when deployed in the real world. Further, L3MVN uses RoBERTa-large, which is smaller than LLaMA-7B in SG-Nav, but it still outperforms SG-Nav on HM3D. ln terms of performance and required computational resources, L3MVN is more suitable to be deployed in real environments than SG-Nav.
> > > >
> > > > I am very grateful to the author for discussing SayPlan in detail. Regarding SayPlan, I should point out that SayPlan also conducts object search tasks (e.g., "Find me something non-vegetarian"), so the comparison is fair if focusing only on that task. In the SayPlan's prompts, they also provide relationships between nodes and edges to represent a hierarchical structure. SayPlan does utilize an already pre-established scene graph, but I don't think this is a significant limitation since scene graph creation can be done in the environment along with the agent. SayPlan can be easily migrated to a mode where the scene graph is created online.
> > > >
> > > > If the authors consider an efficient online 3D scene graph construction method to be a contribution, I suggest the authors should perform ablation experiments on it (e.g., comparing it to other online graph building algorithms) to reflect its contribution and advantage.
> > > >
> > > > I don't deny that the hierarchical chain-of-thought prompting method in SG-Nav works, but I have reservations about whether "designing a CoT prompt" is novel. In recent years, with the development of LLM, CoT techniques have been widely and extensively studied and applied. In many cases, CoT is a default way to use LLM, such as OpenFMNav and SayPlan. Moreover, Prompts in OpenFMNav, with stronger LLM, have better navigation performance. This seems to indicate that OpenFMNav's prompts are more promising compared to structured graph prompts.

---

> > > > > ### Author Response · Authors · 2024-08-13
> > > > >
> > > > > Thanks for the reviewer's timely reply. We answer the questions as below.
> > > > >
> > > > > **1. Navigation performance cannot be improved when LLM reasoning performance is improved**
> > > > >
> > > > > We have mentioned before that we can construct more detailed and larger 3D scene graph to prompt LLM. In this case, the reasoning task for LLM will be harder and thus stronger LLM may perform better. We now conduct experiments to validate this.
> > > > > We relax the condition for edge pruing to increase the number of edges in the scene graph. Then we use this more complex scene graph to prompt LLaMA-7B and GPT-4 respectively. We name this setting as SG-Nav-Complex and run experiments on MP3D. Due to time constraints, we only use 40% of the validation set. As shown below, the performance gap between LLaMA-7B and GPT-4 widen from 0.1 to 0.5 on SR, which means the performance of SG-Nav can be improved by using stronger LLM. In the future, we can construct more detailed 3D scene graph to further exploit the reasoning ability of LLM. We thank the reviewer for this constructive advice and will add this experiments in the revised version of our paper.
> > > > > |  | **SR** | **SPL** |
> > > > > |--|:--:|:--:|
> > > > > |SG-Nav (LLaMA)|40.1|16.0|
> > > > > |SG-Nav (GPT-4)|40.2|16.0|
> > > > > |Gap|**0.1**|**0.0**|
> > > > > |SG-Nav-Complex (LLaMA)|39.3|15.7|
> > > > > |SG-Nav-Complex (GPT-4)|39.8|15.9|
> > > > > |Gap|**0.5**|**0.2**|
> > > > >
> > > > > **2. LLaMA-2-7B and LLaMA-3-8B**
> > > > >
> > > > > We think the reviewer's statement "With these small-volume models with stronger reasoning, the performance of other methods is improved, but not SG-Nav. " may not be exactly correct. To the best of our knowledge, none of the LLM-based ObjectNav methods currently use LLaMA-2-7B or LLaMA-3-8B. L3MVN, VoroNav and OpenFMNav adopt RoBERTa, GPT-3.5 and GPT-4 (GPT-4V) respectively.
> > > > >
> > > > > **3. The performance of L3MVN**
> > > > >
> > > > > We reproduce the performance of L3MVN (Zero-Shot, with RoBERTa-large) on all three benchmarks. As shown below, SG-Nav outperforms L3MVN on five metrics and is only slightly lower in HM3D SR. So generally, we believe the overall performance of SG-Nav is greater than L3MVN. Note that MP3D and RoboTHOR are more challenging than HM3D, which shows SG-Nav is especially superior on complicated and challenging scenarios.
> > > > > |  | **MP3D-SR** | **MP3D-SPL** | **HM3D-SR** | **HM3D-SPL** | **RoboTHOR-SR** | **RoboTHOR-SPL** |
> > > > > |--|:--:|:--:|:--:|:--:|:--:|:--:|
> > > > > |L3MVN|35.5|14.7|**50.4**|23.1|42.2|22.8|
> > > > > |SG-Nav|**40.2**|**16.0**|49.1|**24.3**|**47.5**|**24.0**|
> > > > >
> > > > > **4. Difference with SayPlan**
> > > > >
> > > > > We agree with the reviewer that SayPlan can conduct object search tasks. However, SayPlan is based on a given scene graph. It focuses on task planning, rather than exploration. Even if SayPlan can be adapted to online scene graph is created online, it cannot directly output **where and how to explore the unknown scenes and predict the actions for agent**. Therefore, we think SayPlan can only search objects within the given scene graph because it does not focus on how to explore unseen objects.
> > > > >
> > > > > We think SayPlan and SG-Nav target at different tasks and it is promising to combine them for more complicated embodied tasks. But for ObjectNav, SayPlan cannot replace SG-Nav. We appreciate the reviewers' in-depth insights on SayPlan. We will cite this paper and add detailed comparison in the revised version.
> > > > >
> > > > > **5. Experiments on scene graph construction**
> > > > >
> > > > > We have conducted experiments to validate the effectiveness of our efficient graph construction method in Figure 5 of our paper. In each navigation step, edge generation is the most time-consuming step, which takes 0.11s on average, accounting for 57.5% of the total time. With the proposed method, our SG-Nav is much faster than conventional method like ConceptGraphs.
> > > > >
> > > > > We have also provided some qualitative results of the generated 3D scene graph. Please refer to the attached PDF file in the global rebuttal.
> > > > >
> > > > > **6. The usage of CoT**
> > > > >
> > > > > We do not deny that “In many cases, CoT is a default way to use LLM, such as OpenFMNav”. However, the CoT prompting in SG-Nav is not naïve. Take OpenFMNav for instance, its novelty mainly lies on the usage of several different LLM/VLM to break down the navigation process, rather than design of prompts. While in SG-Nav, we design CoT prompts that make LLM explore the graph structure, which is very different from conventional text-only CoT. Therefore, we believe our method is a simple (but not naive) and effective way for LLM-based 3D scene graph understanding.
> > > > >
> > > > > **In fact, we view the simplicity of CoT prompting as an advantage rather than a shortcoming**. It is easy to implement and achieves satisfactory results on a number of diverse benchmarks. We believe that for exactly that reason our framework is likely to be welcomed by the community. We would also like to note that both reviewer zzdn and aSmS recognize CoT prompting as interesting and well-motivated, which can serve as a starting point for future methods.
> > > > >
> > > > > Thanks again for the constructive comments. We are happy to answer further questions as timely as we can.

---

### Official Review · Reviewer_aSmS · 2024-07-09

**Soundness:** 3
**Presentation:** 4
**Contribution:** 3
**Rating:** 6
**Confidence:** 4

**Summary:**

The paper proposes a 3D scene graph prompting strategy and designs a hierarchical chain-of-thought prompt for improving LLM-based zero-shot object navigation. The 3D scene graph is incrementally updated and pruned to reduce the computational complexity. A re-perception mechanism is also introduced to correct the perception error. Experimental results on MP3D, HM3D, and RoboTHOR environments show the superiority of the proposed method over the competitors.

**Strengths:**

1 The proposal of 3D scene graph prompting and hierarchical chain-of-thought is well-motivated and effective for encouraging the scene context reasoning and improving the decision interpretability.

2 The author provides a detailed time complexity analysis for the proposed edge updating method.

3 Extensive experiments are conducted on three different datasets and the results show the effectiveness of the proposed method.

4 The paper is well-written with figures and tables nicely presented.

**Weaknesses:**

1 The approach utilizes the LLM to predict the possible relationships between objects, and possible distances between object and goal. However, this information may be various in different scenes. What if the prediction is inconsistent with the real scene? Would it largely impact the navigation success? I suggest the author give more results and analyses about this.

2 Although Figure 6 gives the visualization of the navigation process, the author did not provide the visualization for the sequential frontier scoring and reasoning output of LLM. It would be better to present this visualization to help further understand the advantages of the approach.

3 There are some recent works surpassing the reported compared methods, e.g., L3MVN (IROS 2023), VoroNav (ICML 2024), OpenFMNav (NAACL 2024), and VLFM (ICRA 2024). The authors should add them to the table for a more complete comparison and analysis.

**Questions:**

See details in the weaknesses.

**Limitations:**

Limitations have been discussed.

---

> ### Author Rebuttal · Authors · 2024-08-06
>
> **1. Error in the prediction of relationship and distance**
>
> We measure the accuracy of relationship and distance prediction on episodes of MP3D validation set.
> For relationship prediction, we conduct human study to annotate the correctness of each relationship predicted by SG-Nav. For each episode, we let the annotators to evaluate the scene graph of the final navigation step. If all annotators think more than 80% edges in this scene graph correctly describe the real relationship, we annotate this episode with correct relationship.
> For distance prediction, we consider the prediction to be correct if the error between a predicted distance and the actual distance is less than 20%. For each episode, we compute the success rate of distance predictions on the final prompt, because the scene graph at this time is the most complete and the number of subgraphs is sufficient.
> As shown below, the overall accuracy of relationship prediction and distance prediction is **74.7%** and **68.4%** respectively, which is accurate. We observe that correct relationship and distance prediction will increase the success rate of navigation. However, even if the predicted relationship and distance is incorrect, the agent still has a relatively high probability to successfully navigate to the goal (**33.2%** success rate of navigation for incorrect relationship, and **35.4%** for incorrect distance). This validates the robustness of our framework.
>
> SR of navigation:
> | Condition |SR|
> | -- | :--: |
> | Correct Relationship | 42.6 |
> | Incorrect Relationship | **33.2** |
> | Correct Distance     | 42.8 |
> | Incorrect Distance   | **35.4** |
> | Unconditional | 40.2 |
>
> Accuracy of relationship:
> | Result/Relation | Correct Relationship | Incorrect Relationship | Total |
> | -- | :--: | :--: | :--: |
> | Navigation Success | 31.8% | 8.4% | 40.2% |
> | Navigation Failure | 42.9% | 16.9% | 59.8% |
> | Total | **74.7%** | 25.3% | 100% |
>
> Accuracy of distance:
> | Result/Relation | Correct Distance | Incorrect Distance | Total |
> | -- | :--: | :--: | :--: |
> | Navigation Success | 29.3% | 11.2% | 40.5% |
> | Navigation Failure | 39.1% | 20.4% | 59.5% |
> | Total | **68.4%** | 31.6% | 100% |
>
> **2. Visualization for the sequential frontier scoring and reasoning output of LLM**
>
> We provide the visualization in the attached PDF file.
>
> **3. Comparison with more methods**
>
> We further compare SG-Nav with more recent works including L3MVN, VoroNav, VLFM and OpenFMNav on MP3D, HM3D and RoboTHOR. For LLM-based methods (SG-Nav, L3MVN, VoroNav, OpenFMNav), we uniformly set LLaMA-7B as the LLM for fair comparison. The results are shown in the table below. It is shown that SG-Nav achieves state-of-the-art performance on the challenging MP3D and RoboTHOR benchmarks. On HM3D, SG-Nav also outperforms recent methods like L3MVN and VoroNav. However, we observe SG-Nav fall slightly behind VLFM and OpenFMNav on HM3D. This is because HM3D is the relatively simplest benchmark among the three, where SG-Nav’s ability to reason and plan in complex and large scenes cannot be fully utilized. Considering the overall performance on three benchmarks, SG-Nav shows leading advantage, especially on complicated and challenging scenarios.
> | **Method** | **MP3D SR** | **MP3D SPL** | **HM3D SR** | **HM3D SPL** | **RoboTHOR SR** | **RoboTHOR SPL** |
> | -- | :--: | :--: | :--: | :--: | :--: | :--: |
> | L3MVN | 34.9 | 14.5 | 48.7 | 23.0 | 41.2 | 22.5 |
> | VoroNav | 31.2 | 14.2 | 42.0 | 26.0 | 38.4 | 22.2 |
> | VLFM | 36.2 | 15.9 | 52.4 | **30.3** | 42.3 | 23.0 |
> | OpenFMNav | 37.2 | 15.7 | **52.5** | 24.1 | 44.1 | 23.3 |
> | **SG-Nav** | **40.2** | **16.0** | 49.1 | 24.3 | **47.5** | **24.0** |

---

> > ### Comment · Reviewer_aSmS · 2024-08-12
> >
> > Thanks to the authors for providing additional quantitative and qualitative results. Most of my concerns are addressed and I am happy to raise my score. I sincerely suggest the authors add these results and analysis to their revised versions, which are helpful for improving the paper further.

---

> > > ### Author Response · Authors · 2024-08-12
> > >
> > > We appreciate the reviewer for the positive feedback. Your constructive comments and suggestions are indeed helpful for improving the paper. Also, many thanks for raising the score.
> > >
> > > We will continue to improve our work and release the code. If the reviewer has any follow-up questions, we are happy to discuss!

---

### Official Review · Reviewer_7ZKj · 2024-07-12

**Soundness:** 3
**Presentation:** 3
**Contribution:** 3
**Rating:** 6
**Confidence:** 3

**Summary:**

This paper propose to use 3D Scene Graph Prompt in LLM-based Zero-shot Object Navigation, which fully use information of whole scene and is explainable. Also it propose prune-based method to accelerate the construct of graph. Enough expriments show the superority of the method and the effectiveness of each modules.

**Strengths:**

1.This paper solves the problem of insufficient use of scene information in LLM based Zero shot Object Navigation in the past.

2. A fast way to establish a scene graph is proposed to ensure fast speed, and theoretical proof is provided.

3. The experiment achieves good results and achieved SOTA results on multiple datasets.

4. The language of the paper is fluent, easy to understand, and there are few grammar errors.

**Weaknesses:**

1.VLM and is needed for Short Edge Pruning which may increase the time consumption of model to construct graph.

2.The Incremental Updating and Pruning may have some issue. (See in Question.)

**Questions:**

1.In line 39-41, the author mentions previous methos are abstract and unexplainable. Please explain this viewpoint in more detail and show how your method solve this problem.

2.Will the already constructed scene graph vary based on changes in perspective? (For example, in the previous timestamp when A and B objects did not appear in the same image, Long Edge Pruning is required, while in the next time when A and B appear in the same image, Short Edge Pruning is required, the scene graph may be differnet). Does using Incremental Updating and Pruning ignore this change and result in cumulative errors.

---

> ### Author Rebuttal · Authors · 2024-08-06
>
> **1. Inference latency of SG-Nav**
>
> In each step of navigation, the time cost of SG-Nav can be divided into perception, graph construction and reasoning, which takes 0.3s, 1.3s, 0.14s on average, accounting for 17.3%, 74.8% and 7.9%. The edge pruning belongs to graph construction, which only takes **11.5%** of the overall time.
> | **Component**             | **Average Time (s)** | **Percentage (%)** |
> |---------------------------|:----------------------:|:--------------------:|
> | Perception                | 0.3                  | 17.3               |
> | Graph Construction        | 1.3                  | 74.8               |
> | &emsp;&emsp;**Edge Pruning**  | **0.2** (part of GC)      | **11.5** (part of GC)       |
> | Reasoning                 | 0.14                 | 7.9                |
>
> **2. Why SG-Nav is explainable compared with previous methods like L3MVN and ESC**
>
> Previous LLM-based ObjNav methods like L3MVN and ESC directly score each frontier based on the nearby object categories. For example, if there are table, computer and chair near a frontier, these method directly prompting language model with the text of these three categories to predict the probability of this frontier, which do not consider the relationship between objects.
>
> **3. Inconsistency of Long Edge and Short Edge**
>
> We thank the reviewer for the constructive advice. Currently, we do not consider the change of perspective in our paper. We further add this verification mechanism to our system. At each timestamp, we not only validate newly generated edges, but also check whether there are some existing object nodes that has been updated (merged with newly detected objects). For edges which have at least one updated node, we also perform validation on them. With this modification, the performance of SG-Nav changes from 40.2/16.0 to 40.2/16.1 (SR/SPL), which shows that the proportion of inconsistency between long and short edges in actual samples is relatively small.
> | **Method** | **SR** | **SPL** |
> | -- | :--: | :--: |
> | w/o Verification | 40.2 | 16.0 |
> | **w Verification** | **40.2** | **16.1** |

---

> > ### Comment · Reviewer_7ZKj · 2024-08-13
> >
> > Thanks for the authors' further clarification. The response has resolved my confusion, so I decide to keep the score of 6.

---

> > > ### Author Response · Authors · 2024-08-13
> > >
> > > We appreciate the reviewer for the positive feedback. Your constructive comments and suggestions are indeed helpful for improving the paper.
> > >
> > > We will continue to improve our work and release the code. If the reviewer has any follow-up questions, we are happy to discuss!

---

### Official Review · Reviewer_zzdn · 2024-07-12

**Soundness:** 3
**Presentation:** 3
**Contribution:** 3
**Rating:** 6
**Confidence:** 5

**Summary:**

This paper introduces a zero-shot object navigation framework using a 3D scene graph to represent the environment. It employs a hierarchical chain-of-thought prompt to LLMs for goal location navigation and includes a re-perception mechanism to correct errors. Experiments are conducted on MP3D, HM3D, and RoboTHOR.

**Strengths:**

- The task is aimed at open vocabulary ObjectNav, which is a more general task.

- Edge updating considers computational complexity.

- The idea of a hierarchical chain-of-thought method to prompt LLM is interesting.

- The paper is well-structured and easy to understand.

**Weaknesses:**

- Although the paper claims to target open vocabulary (zero-shot) object navigation, the method uses some closed-set settings, such as a predefined relationship dictionary, which contradicts the open vocabulary premise.

- In the experiments, the target objects chosen (on MP3D and HM3D) are the same as those in regular ObjectNav settings, where the objects are common. This seems to undermine the method's capability in handling open vocabulary objects.

- The proposed 3D scene graph is defined with objects, groups, and rooms. There is a related work for ObjectNav that also constructs a similar graph with objects, zones, and scene nodes. I suggest that authors can compare their graph construction with this work and include it in the references to enrich the paper.
[1] Hierarchical object-to-zone graph for object navigation, ICCV 21.

**Questions:**

- How is the accuracy of the 3D scene graph validated?

- The chain-of-thought method to prompt LLM is interesting. Are there other methods to prompt LLM? Comparing different prompt methods could make the experimental results more solid.

- Is the zero-shot object definition meant to be training-free, or does it mean that the target is not predefined? While the problem definition mentions open-vocabulary, Figure 1 and the selected object categories in the experiments are still common objects.

- I would like to see some failure cases. What are the reasons behind these failures?

**Limitations:**

The authors discuss potential limitations in the conclusion section.

---

> ### Author Rebuttal · Authors · 2024-08-06
>
> **1. About open-vocabulary setting, is it training-free**
>
> Our method is training-free and open-vocabulary. Since most objects occur in the scene belong to some common categories, we can pre-define the relationship among these categories and save them to a dictionary, which accelerates inference speed. Note that the relationship is acquired by prompting LLM, so this process is also training-free. When SG-Nav receives object categories that have not been saved before, we simply prompt LLM to generate the relationship and update the dictionary.
>
> **2. The choice of object categories**
>
> We select the categories as same as those in regular ObjectNav settings in order to compare with supervised and close-vocabulary methods, following the experimental setting of ESC. But SG-Nav is not limited to any category set.
>
> **3. Comparison with “Hierarchical object-to-zone graph for object navigation”**
>
> This paper proposes a hierarchical object-to-zone (HOZ) graph for ObjectNav. There are several differences between SG-Nav and this work. First, SG-Nav is completely training-free, but HOZ needs to finetune a Fast-RCNN for graph construction and train a LSTM policy module via reinforcement learning. Second, the 3D scene graph in SG-Nav contains more hierarchy and relationship between objects, while HOZ mainly consists of zone nodes and using adjacent probability as the edges between nodes. Third, SG-Nav adopts LLM to exploit the hierarchical and structural information in 3D scene graph in zero-shot, while HOZ trains a network to parse the graph. We will cite this paper and compare with it in the final version of paper.
>
> **4. The accuracy of 3D scene graph**
>
> Since our method mainly focuses on how to leverage 3D scene graph to enable context-aware reasoning for ObjectNav, we do not quantitively evaluate the accuracy of 3D scene graph. To better demonstrate the intermediate process of SG-Nav, we provide some visualization results in the attached PDF file.
>
> **5. Ablation on CoT-Prompting**
>
> We further design ablation study on the Chain-of-Thought Prompting method. The raw performance on MP3D is 40.2/16.0 (SR/SPL). If we directly convert the 3D scene graph into a text and prompting LLM, the performance on MP3D is 36.5/14.9 (SR/SPL). By converting the nodes and edges into text separately, the performance on MP3D is 37.0/15.0 (SR/SPL), which is better than previous one but still worse than our CoT-prompting. The results show that fully exploiting the hierarchical graph structure of scene graph can help LLM understand the scene context and make better decision.
> | **Method** | **SR** | **SPL** |
> | -- | :--: | :--: |
> | Text prompting | 36.5 | 14.9 |
> | Text prompting seperately | 37.0 | 15.0 |
> | **Raw** | **40.2** | **16.0** |
>
> **6. Failure case**
>
> We demonstrate failure case in the attached PDF file. Please refer to the global rebuttal.

---

> > ### Comment · Reviewer_zzdn · 2024-08-12
> >
> > Many thanks for the detailed rebuttal. The responses have addressed some of the concerns.
> >
> > In the response, the authors mentioned, "When SG-Nav receives object categories that have not been saved before, we simply prompt LLM to generate the relationship and update the dictionary." Does this imply that the proposed method mainly benefits pre-defined categories? If so, although the method can accommodate open vocabulary, the actual focus of the paper still revolves around pre-defined categories. This limits the method's generalizability, especially since it aims at zero-shot object navigation.

---

> > > ### Author Response · Authors · 2024-08-12
> > >
> > > Thanks for the reply. We explain the question as below:
> > >
> > > The purpose of pre-defining a dictionary is **only to improve navigation efficiency**, which is **irrelevant to the performance**. Because we can save the relationships between common objects into the dictionary to save the time for LLM prediction.
> > >
> > > Even if we do not pre-define any category and start with an empty dictionary, the performance of SG-Nav will not change. In this way, it will be slower in the early episodes. But as more and more scenes are explored, the dictionary will gradually accumulate and cover common categories, and then SG-Nav will become faster. So the role of the pre-defined dictionary is to skip the early episodes and make SG-Nav efficient even at the beginning. It **does not affect the zero-shot ability** of SG-Nav.

---

> > > > ### Comment · Reviewer_zzdn · 2024-08-14
> > > >
> > > > Thanks for the response. The reply addresses my concerns regarding whether the proposed method benefits only pre-defined objects. Therefore, I decide to maintain my previous rating.
> > > >
> > > > Additionally, considering that this work focuses on an open-vocabulary object setting, I understand the authors' choice to use a consistent task setting for fair comparison with previous work. However, I suggest including evaluations on uncommon object categories in the final version to further strengthen the paper.

---

> > > > > ### Author Response · Authors · 2024-08-14
> > > > >
> > > > > We appreciate the reviewer for the positive feedback. We will add experiments on uncommon categories for a more comprehensive evaluation.
> > > > >
> > > > > We will also continue to improve our work and release the code. If the reviewer has any follow-up questions, we are happy to discuss!

---

### Author Rebuttal · Authors · 2024-08-06

We would like to thank the reviewers for their valuable and constructive comments. We provide detailed answers to these questions and will revise the paper accordingly.

---

### Decision · Program_Chairs · 2024-09-25

**Decision:**

Accept (poster)

**Comment:**

This paper presents a zero-shot object navigation framework with a 3D scene graph to represent the observed scene.
Initially it received rating of 6654.
Most of the concerns were resolved during rebuttal and the Reviewer aSmS raised the score from rating of 5 to 6.
However, the Reviewer JsEs found the performace and novelty are not strong enough, and the rating was dropped from 4 to 3 after rebuttal.
The author provided detail response and additional experiments.
The AC carefully looked into this case and had discussion with the SAC.
Both the SAC and AC agree that the requested experiment results are qualified to support the argument of this paper, and the concerns from Reviewer JsEs are not significant enough to reject the paper.
The AC thus recommends acceptance for this paper.
The authors are urged to include the new experiments (e.g., ones with stronger LLMs) in the camera-ready version of the paper.